# Analytical Characterization of the Intercalation of Neutral Molecules into Saponite

**DOI:** 10.3390/molecules27103048

**Published:** 2022-05-10

**Authors:** Valentina Toson, Diego Antonioli, Enrico Boccaleri, Marco Milanesio, Valentina Gianotti, Eleonora Conterosito

**Affiliations:** 1Dipartimento di Scienze e Innovazione Tecnologica, Università del Piemonte Orientale, 15121 Alessandria, Italy; valentina.toson@uniupo.it (V.T.); diego.antonioli@uniupo.it (D.A.); marco.milanesio@uniupo.it (M.M.); 2Dipartimento per lo Sviluppo Sostenibile e la Transizione Ecologica, Università del Piemonte Orientale, 13100 Vercelli, Italy; enrico.boccaleri@uniupo.it (E.B.); valentina.gianotti@uniupo.it (V.G.)

**Keywords:** saponite, layered material characterization, TGA-GC-MS, fluorescence spectroscopy, intercalation evaluation

## Abstract

Organo-modified layered materials characterization poses challenges due to their complexity and how other aspects such as contamination, preparation methods and degree of intercalation influence the properties of these materials. Consequently, a deep understanding of their interlayer organization is of utmost importance to optimize their applications. These materials can in fact improve the stability of photoactive molecules through intercalation, avoiding the quenching of their emission at the solid state, to facilitate their use in sensors or other devices. Two synthetic methods for the preparation of saponites with a cationic surfactant (CTABr) and a neutral chromophore (Fluorene) were tested and the obtained products were initially characterized with several complementary techniques (XRPD, SEM, TGA, IR, UV-Vis, Fluorescence and Raman spectroscopy), but a clear understanding of the organization of the guest molecules in the material could not be obtained by these techniques alone. This information was obtained only by thermogravimetry coupled with gas chromatography and mass spectroscopy (TGA-GC-MS) which allowed identifying the species present in the sample and the kind of interaction with the host by distinguishing between intercalated and adsorbed on the surface.

## 1. Introduction

In recent years, increasing interest has been manifested in the synthesis of organo-clays hybrid materials. In fact, such materials combine the advantages of organic functionalities and inorganic frameworks stability in a unique solid. These materials can be in fact applied to improve the stability of photoactive molecules through intercalation and to avoid the quenching of their emission at the solid state. Moreover, the co-presence of different functionalities in the same material may result in peculiar synergistic effects that can be exploited for several applications such as catalysis, drug delivery, additives and optoelectronics [1,2,3,4,5,6]. Moreover, the possibility to intercalate substances that are in the liquid state at RT can solve engineering issues caused by having liquids inside a device, such as dye-sensitized solar cells or batteries, and they can also act as a dispersant or change the surface characteristic of the intercalated compound to facilitate their use as additives in polymers [2]. On one hand, the main advantage in the application of natural clays is their large availability and low price. On the other hand, technological applications often require highly pure materials of synthetic origin, obtainable by time and resource-consuming processes. To obtain clay-like materials with a controlled chemical composition, several methods have been developed [5,7,8,9,10,11,12,13,14,15,16].

Saponite is a trioctahedral 2:1 clay mineral belonging to the smectite group, with chemical formula: M^n+^_x/n_[Mg]_6_(OH)_4_[Si_8−x_Al_x_]O_20_∙mH_2_O, able to host organic/inorganic cationic species. Organo-saponites are generally prepared by a post-synthesis ion exchange approach from natural or synthetic inorganic saponites. The ion exchange process is usually carried out in solution with large consumption of solvents. A viable and more effective procedure is represented by almost dry techniques such as Liquid Assisted Grinding (LAG) [2,17,18,19], able to drastically reduce chemical consumption. To further move towards green and facile preparation methods, one-pot strategies have been developed for a single step preparation of organo-clays in general, and organo-saponite in particular [20]. These methods have several advantages such as the reaction speed-up and a homogeneous distribution of molecules in the lamellae [21,22] but the organic moieties undergo thermal and chemical stress caused by the hydrothermal conditions.

A very interesting family of organo-modified layered materials can be obtained by the intercalation of chromophores to prepare nanostructured organo-clays materials with fine-tuned optical properties [2,23]. The dispersion of the chromophore in the inorganic guest can, in fact, reduce self-absorption and quenching effects, because the host–guest interactions can affect the distribution and orientation of guests in the host. Moreover, the intercalation can also lead to thermal and photochemical stabilization of the guest. 

Since only cationic chromophores can be introduced into the negative saponite lamellae, special strategies are needed to intercalate neutral molecules. One approach, proposed for hydrotalcites [24], is the interlayer confinement of chromophores by their dispersion in surfactants. This approach has the additional advantage of controlling the average distance between the photoactive species and hence their interactions.

In this work, two strategies to prepare a fully synthetic organo-modified saponite hosting neutral molecules were adopted, namely the classical one-pot (OP) and, for the first time on a cationic layered material, the Liquid Assisted Grinding (LAG) method. 9H-Fluorene (Fluorene) was chosen as the model of a neutral fluorescent molecule and Cetyltrimhetylammonium Bromide (CTABr) was the surfactant employed as a vehiculant to obtain the intercalation of the guest (Figure 1). 

To fine-tune the properties, the materials obtained must guarantee a high degree of purity and maintain the stability of both the host and the guest. Indeed, many aspects such as contamination, preparation methods and degree of intercalation influence the properties of such materials, and many inorganic and organic-exchanged saponites show defectivity, presence of organic adsorbed on the material surface and at defects, or degradation. Fluorene and CTABr successful intercalation was investigated by XRPD, FTIR, Raman Spectroscopy, and Diffuse Reflectance UV-Visible and Fluorescence spectroscopy. Thermogravimetric analysis under oxidative conditions allowed the evaluation of the thermal stabilization effect of the organic molecule after intercalation into saponite. Finally, SEM analysis was performed to evaluate the effects of intercalation methods and of the organic guests on saponite morphology. 

These classical techniques are also able to suggest the presence or absence of the abovementioned defects and purity issues but not always to demonstrate them clearly and quantitatively.

A promising technique to overcome these limitations appears to be thermogravimetry coupled with chromatographic separation and mass spectrometric detection (TGA-GC-MS). It was successfully employed to characterize organo-modified anionic layered samples [25]. In fact, the addition of a Gas Chromatographic (GC) stage between TGA and MS was effective in improving the analytical performance for contamination evaluation at the nano-scale level. In this way, the TGA can operate to impart a precise and reproducible thermal history to the samples and the hyphenation of TGA with GC-MS allows the separation of the evolved gas mixture into single components [26] with a much easier and resolved MS information and an elevate sensitivity [27,28]. TGA-GC-MS was also able to provide information about the strength of interactions between the inorganic skeleton of saponite and the organic molecules through the analysis of the thermal behavior of the organic molecules.

## 2. Results

Fluorene-containing saponite materials were obtained by two methods, OP and LAG as detailed described in the experimental section. These methods were adapted and tuned starting from literature procedures with the aim of obtaining the complete exchange and intercalation of guest molecules and limiting the amount of waste solutions [24,29,30].

For the LAG method, tests were made with different concentrations of HCl in the solution, which at last proved to be unnecessary. The amount of ethanol used was also optimized to be the minimum necessary to reduce waste. The amount of CTABr was increased up to four times higher than the cation exchange capacity (CEC) of the saponite to maximize the amount of intercalated CTABr. 

While the OP approach allows to obtain in a single step the introduction of the surfactant and the neutral molecule in the saponite during the hydrothermal synthesis of the saponite itself, the LAG one, originally developed and optimized by Milanesio et coworkers [17,18,19] for Layered Double Hydroxides (LDH), should allow intercalation of the molecules of interest in an already synthesized sodium saponite (Na-SAP110) at room temperature with a solvent-free reaction. 

The characterization was performed, at first, by the classical techniques for morphological, thermal and photophysical evidence, then TGA-GC-MS analysis was exploited to shed light on the species, environment and organization inside the materials. 

### 2.1. Structural and Morphological Characterization

The first information required for the characterization of the synthesized materials is, of course, the success of the intercalation. This can be easily demonstrated by XRPD by looking at the increase of the interlayer spacing, revealed by a shift of the basal peaks of layered materials towards low angles.

Figure 2, reporting the XRPD patterns of Na-SAP110, CTA_FL_SAP_OP and CTA_FL_SAP_LAG and CTA_SAP_LAG for comparison, shows clearly that CTABr has entered (thanks to the LAG process) the layers in its cationic form CTA^+^ (operating on a batch of Na-SAP110), as confirmed by the increase of the basal spacing (001) in 2θ from 7.88° (11.2 Å) of the inorganic Na-SAP110 sample (curve a), to 6.1° (14.4 Å) in the LAG samples (curve b and c). Curves b and c allow comparing the LAG procedure with and without fluorene in the LAG conditions. It can be stated that the interlayer spacing, which is basically related to the cationic surfactant, is not affected by the presence of the neutral organic molecule.

Again, from Figure 2 it is clear that the OP method, despite using a mixture also containing fluorene, leads to a saponite layered material consistent with the literature reference. Notably, the basal spacing (001) in 2θ at 7.88° (11.21 Å) of the inorganic Na-SAP110 sample (curve a), moves to 6.36° (13.9 Å) in the OP sample (curve d).

So far, fluorene molecules are expected to be intercalated in between the CTA^+^ alkyl chains and therefore are able to affect the interlayer spacing directly.

Overall, the presence of CTABr and fluorene does not change the characteristic trioctahedral structure of saponite, as shown by the position of the reflection due to the (060) plane at 60.6° in 2θ.

The difference of 0.5 Å between the interlayer distance of CTA_FL_SAP_OP and CTA_FL_SAP_LAG could be ascribed to a different organization of CTA^+^ chains related to the preparation method. 

A larger crystallinity can be evidenced in both the intercalated samples but is more evident in the one-pot sample (Figure 2 curve c). The increase of crystallinity and the shorter interlayer distance agree with a flat monolayer of CTA^+^ intercalated with its alkyl chains, parallel to the inorganic layer and this can be further supported by the FTIR data reported in Figure 3. It is noteworthy that the IR spectra mainly show the CTA^+^ features as the spectral contribution of fluorene is less relevant. In the FTIR spectra, the regions of the asymmetric stretching and bending modes of CH_2_ (2800–3000 cm^−1^ and 1400–1520 cm^−1^, respectively) are very rich in information. In fact, the frequency and the width of the bands in those regions are sensitive to the gauche/trans conformer ratio and to interchain interactions between CTABr molecules [31,32,33,34,35]. In detail, in both CTA_FL_SAP_OP and CTA_FL_SAP_LAG, a contemporaneous shift of the asymmetric (~2920 cm^−1^) and symmetric (~2850 cm^−1^) stretching bands to a higher frequency with respect to CTABr, can be ascribed to the predominance of the gauche conformer indicating that the chains are in a liquid-like state.

Relevant changes can also be seen in the δ(CH_2_) scissoring and δ_as_(N–CH_3_) mode, falling at 1462 and 1472 cm^−1^ and 1480 and 1487 cm^−1^, respectively. When the δ(CH_2_) band is observed at 1472 cm^−1^, it indicates the high organization of the CH_2_ chain conformation (this is evident in pure CTABr but lost in all the intercalated materials). In addition, a signature related to the lack of chain–chain ordering is the split of the δ(CH_2_) band, which is mainly due to the lateral interchain interaction between CH_2_ groups of adjacent chains. This splitting is absent in alkane chain assemblies when lateral interchain interactions are weak. 

The OP sample, conversely, shows, in the δ(CH_2_) region, mainly one peak, consistent with the complete lack of organization of the CH_2_ chain conformation. Quite surprisingly, the FTIR spectra of the physical mixture (Phys-Mix) sample are different from the mere sum of the CTABr and Na-SAP110 samples spectra, especially in the 1500–450 cm^−1^ region, suggesting a loss of interchain interactions and a possible beginning of intercalation.

SEM analyses were performed on the intercalated samples and are reported for brevity reasons in ESI materials (Appendix A). They confirmed that both synthetic methods produced saponite materials with the characteristic layered morphology. The OP sample is poorly delaminated and shows the presence of aggregates of packed layers of a few microns thickness. Conversely, a larger delamination degree and a lower dimension of the aggregates can be observed in the LAG sample. 

This is probably due to the fact that CTA^+^ ions have an organizing effect on the layers, as also already reported in the literature [36], in fact, the XRPD pattern of the intercalated materials shows a sharper basal peak with respect to Na-SAP110, indicating a more constant spacing between the layers. In OP synthesis the sheets are formed simultaneously to the packing of the layers with CTA^+^ therefore the material is more ordered. In the LAG synthesis, CTABr has instead to enter the layers of an already formed and disordered saponite and therefore the re-ordering is less effective, so in this case, the crystallinity of the material and the morphology are linked.

### 2.2. Photophysical Characterization

Photophysical characterization was performed to obtain an insight into the arrangement of fluorene. The effective presence of fluorene in the interlayers was investigated by registering the DR-UV reflectance spectra reported in Figure 4. The UV/Vis and emission spectra of CTABr can be found in the literature [37] The UV/Vis spectra of fluorene and of Na-Sap110 are reported in Appendix A.

The absorption spectrum shapes of CTA_FL_SAP_OP and CTA_Fl_Sap_LAG are quite similar and the characteristic fluorene absorptions in the ultraviolet region (see Appendix A) are present. The intensities of absorption are instead very different, as the absorption of the OP sample was 60% less than with respect to the LAG one. This can be due to a different amount of fluorene in the sample or to a different arrangement in the interlayer. 

Unexpectedly, the CTA_FL_SAP_OP spectrum shows an absorption band in the visible region at about 530 nm which may be due to the presence of some by-products formed in the harsher synthetic conditions employed in the OP method (i.e., non-ambient temperature and pressure, basic reaction environment, presence of acid catalytic sites on the saponite substrate).

Fluorescence emission spectra, obtained irradiating at λ = 260 nm, are reported in Figure 5A and they show a broad emission band centered at 314 nm, with intensities in agreement with the adsorption profiles. The LAG sample emission is more intense than that of the OP sample, probably due to the better dispersion of fluorene among the CTABr chains, reducing the quenching effects from the interaction between fluorene molecules or due to a larger amount of fluorene.

Given the presence of the absorption band located at 530 nm in the CTA_FL_SAP_OP spectrum (Figure 4), an emission spectrum was also recorded with irradiation at λ = 530 nm and is reported in Figure 5B together with spectra obtained with irradiation at 260 nm recorded using the same instrumental setup to allow comparison. The emission, when exciting at 530 nm, is stronger compared to the one at 260 nm. The presence of this strong emission could mean the formation of fluorene aggregates emitting at a higher wavelength [38] or the formation of fluorene by-products at the expense of the amount of fluorene emitting at 314nm. These hypotheses were further investigated, and the discussion is reported in paragraph 2.5.

### 2.3. Thermogravimetric Analysis in Oxidizing Conditions

The TGA curves collected under airflow conditions of Na-SAP110 and of the two organo-modified samples are reported in Figure 6 together with the differential TG curves (dTG). 

The thermogravimetric profile of Na-SAP110 shows two weight losses. The first, centered at 120 °C corresponds to physisorbed and chemisorbed water, and the second one is centered at 490 °C to de-hydroxylation. 

Na-SAP110 contains about 7% of physisorbed and chemisorbed water (calculated from the extent of the losses at 120 °C), while CTA_FL_SAP_OP, CTA_FL_SAP_LAG show both a lower extent of water (with a weight loss of 1.9% and 2.6%, respectively) and a lower temperature of evolution (around 100 °C according to dTG). Hence, the organo-modification of saponite with CTABr seems to change the hydrophilic behavior of the saponite interlayer reducing the amount and the strength of interaction of water with the host structure. 

In the organo-modified samples, the main weight loss occurs between 240 and 400 °C. In the LAG sample, the higher weight loss, of about 8.6%, appears to be due to different processes occurring in a broad range of temperatures that can be interpreted as the initial loss of CTABr outside the layers or poorly interacting (centered at 300 °C) and the subsequent loss of intercalated CTA^+^. In the OP sample, the loss of the organic fraction is about 7.5% in weight and occurs at higher temperatures, between 300 and 400 °C and in a single step indicating that CTA^+^ is better stabilized and evenly distributed inside the layers. 

The dehydroxylation weight loss centered at 490 °C of saponite can be estimated from the Na-SAP110 analysis and it is equal to 1.5%. Hence, the remaining weight loss is the combustion of the organic molecules. Moreover, the two organo-saponite samples show the formation of a carbonaceous phase, due to char formation promoted by the acid sites of saponite, which is removed at temperatures higher than 600 °C. 

The presence of both fluorene and CTABr is thus inferred by these data, but the superposition of weight losses due to the structure of saponite and organic moieties makes the precise quantification of the organic moieties difficult. This issue is overcome in the TGA-GC-MS analysis where the different contributions are resolved by the GC stage and identified in the MS stage, giving the bullet proof of fluoere insertion within the layers. 

### 2.4. TGA-GC-MS Analysis

With the aim of elucidating the reasons behind the different behavior of the saponites obtained by OP and LAG synthesis TGA-GC-MS analyses were performed on all the materials and on the single components.

Moreover, the physical mixture material was prepared (as reported in the experimental section) and analyzed. 

In fact, a gas chromatographic (GC) separation stage between the TGA and the mass spectrometer detector (MS) allowed the separation of the gas mixture evolved from TGA into single components and their identification by highly-resolved MS data.

An important difference in this technique is also related to the use of He as a fully inert carrier gas, that allows the determination of the released species according to their molecular nature or the thermal degradation compounds, without the effects of oxygen as in TGA data shown above.

The mass spectrometric acquisitions were performed both in full-scan mode, in the 20–350 *m*/*z* range and in Single Ion Monitoring (SIM) mode by acquiring the signals characterizing the organic molecules under study. In detail, several scouting analyses performed on the pure components allowed to identify the fragment at *m*/*z* = 95 that can be used to monitor the CTABr and the fragment at *m/z* = 166 was employed to follow the fluorene evolution profile. By comparing the temperatures and the shape of the evolution profile of each species it is possible to infer the environment of the compound in the intercalation compounds. In fact, the pure guests usually have a narrow signal centered at a lower temperature with respect to the intercalated ones which are evolved at higher temperatures. A wide weight loss with two distinct maxima demonstrates different chemical environments, sugegsting that some of the guest is outside the layers and therefore is not stabilized, while some other is inside the layers and therefore evolved at higher temperatures [25,39,40].

As can be seen in Figure 7A,B, the lower part, the dTA of CTABr and Fluorene (dotted lines) are perfectly described by the evolution profiles of the chosen fragments (95 *m*/*z* for CTABr and 166 *m*/*z* for fluorene) which show a single broad loss, extending from 120 to 200 °C for fluorene and from 200 to 300 °C for CTABr. 

Regarding the CTABr signal, moving from top to bottom in Figure 5, it can be observed that both the OP and LAG materials present this signal when the main weight loss occurs. The maximum of the evolving peak is at a higher temperature in the case of the OP sample (about 400 °C) while the release is more spread at lower temperatures in the case of the LAG sample. It is interesting to observe the profile of the physical mixture sample which has two evolving peaks, one approximately corresponding to the pure CTABr (T of about 230 °C) and one with the maximum at the same temperature that is found in the LAG and OP materials (T of about 380 °C). These observations allow us to conclude that intercalation of CTABr occurs, and it causes a thermal stabilization of this surfactant in the two saponites. Moreover, in the physical mixture, the TGA-GC-MS analysis proved to be capable of discriminating between the CTABr adsorbed or weakly bound on the material and the intercalated one that results as stabilized, confirming that CTABr can interact with saponite even with a simple mechanical action (as already suggested by FTIR data).

Even observing the typical fragment of fluorene at *m*/*z* 166 (Figure 7B), it can be noted that, already in the physical mixture, there are two evolution peaks corresponding to the weight losses centered at 200 °C and at 400 °C, indicating the simultaneous presence of fluorene adsorbed and more strongly bonded. The same trend can be observed in the two intercalated materials. The less marked profile is due to the lower sensitivity for the fluorene fragment detection in mass spectrometry with respect to that of the CTABr, as can be deduced by observing the intensities of the evolution profiles of the pure compounds that have been acquired using the same conditions.

These observations demonstrated that an important portion of the fluorene used during the preparation procedure of the materials was intercalated with both the OP and LAG methods. As a comparison, observing the 166 *m*/*z* profile for Na-SAP110, no peaks or bands are evident and only a drift due to the background corresponding to the temperature increase of the TGA thermal gradient is observed.

### 2.5. By-Product Formation Investigation

Since the above analyses suggested a modification of the fluorene incorporated by the OP technique, this material has been further investigated. In fact, the harsh experimental conditions and the contemporaneous presence of aluminum isopropoxide, alcohol isopropyl, catalytic substrate (saponite) and basic environment (NaOH) are known to be able to promote fluorene alkylation and/or the formation of fluorenone which can actually take place under even milder oxidation conditions [41,42,43,44]. Possible fragments deriving from fluorene modification were searched for in the TGA-GC-MS profiles acquired in the full-scan mode of the sample CTA_FL_SAP_OP. 

As can be seen in the lower part of Figure 8, a signal deriving from the fragment at 180 *m*/*z* was found and the relative peak spectrum confirms, with a probability of 70%, that can be consistent with fluorenone molecules when compared with the Wiley spectra library.

To deeply investigate the presence of fluorene derivates, OP saponite was subjected to a solvent extraction via toluene followed by a water washing to dismiss the CTABr eventually extracted. After evaporation, a yellow slurry (Ext) was obtained. Both the obtained extract and the remaining post-extraction saponite material (CTA_FL_SAP_OP_ext) were characterized. The TGA-GC-MS profiles are reported in the lower part of Figure 8 for what regards the mass fragment of fluorenone (180 *m*/*z*) while at the top of the same figure the chromatograms of the CTABr (*m*/*z* 95) and fluorene (*m*/*z* 166) are reported, respectively.

From such figures, it can be observed that the toluene extraction well performs with fluorene and fluorenone but it extracts very little of the CTABr (see the sample Ext chromatogram in each figure). 

From the evolution profile obtained for the post-extraction saponite (CTA_FL_SAP_OP_ext), it can be observed that, as expected, the CTABr peak is still present in a high amount, while only a little amount of fluorenone (180 *m*/*z*) and fluorene remains intercalated. The fact that some fluorene could not be removed by the extraction procedure is another hint that intercalation was successful and the different evolution temperatures of the fluorene peaks in the extract with respect to the CTA_FL_SAP_OP_ext confirms once again that the molecule is intercalated and strongly stabilized. 

On the other hand, no fragments of the alkyl derivatives of fluorene hypothesized above were found as can be seen in Appendix A in which, as examples, fragments 250 *m*/*z* for di-alkylated and 208 *m*/*z* for mono-alkylated fluorene are reported.

Moreover, these indications are further confirmed by the analyses carried out using other techniques. 

The optical features of the saponite after the extraction were investigated by DR-UV reflectance and fluorescence spectroscopy. The normalized absorption profiles of CTA_FL_SAP_OP_ext and CTA_FL_SAP_OP differ for both the number of absorptions and their intensities, as shown in Figure 9. After the extraction, the visible absorption at around 530 nm disappears (see CTA_FL_SAP_OP_ext spectrum in Figure 9), confirming the possible presence of a chromophore fluorene by-product in the interlayer of saponite. Only a trace of the most intense signal of fluorene at 265 nm can be seen.

It is worth noting, that the host structure shows no structural differences after the extraction treatment (Figure 10A) as demonstrated by the superposition of the XRPD patterns of CTA_FL_SAP_OP and CTA_FL_SAP_OP_ext. However, the extraction success is demonstrated by comparing the FT-Raman spectra of the extracted fraction with the material before and after the extraction.

Characteristic modes of saponite structure (marked as - in Figure 10B) and of CTABr (marked as * in Figure 10B) are still present after the toluene extraction in CTA_FL_SAP_OP_ext whereas, despite the evident fluorescent background, the characteristic modes of fluorene, enriched by signals highlighting the presence of other by-products, are found in the spectrum of the extract, aside a residual amount of surfactant despite the water extraction step. 

More detailed information can be obtained from the FTIR spectra (Figure 10C). It clearly shows the main component of the extract fraction is fluorene but as IR patterns are helpful in identifying the by-product functionalities, it appears evident that the presence of a ketonic functional group, addressable to fluorenone, is in agreement with the band at 1716 cm^−1^, that is evident also in the Raman spectra.

## 3. Materials and Methods

### 3.1. Materials

All chemicals were purchased by Sigma Aldrich (now Merck KGaA, Darmstadt, Germany) analytical grade and employed without further purification. 

### 3.2. Saponite Synthesis

A synthetic sodium saponite sample with a H_2_O/Si ratio equal to 110 (Na-SAP110) was prepared by hydrothermal synthesis starting from a gel with the composition of 1SiO2:0.835MgO:0.056Al_2_O_3_:0.056Na_2_O:110H_2_O accordingly to a procedure already reported in the literature [45]. This material was used as reference (Na-SAP110) and as starting material for LAG intercalation. 

### 3.3. LAG Synthesis

Fluorene (47.8 mg) and CTABr (253.4 mg) in molar ratio 0.4:1, were mixed and ground at 1200 RPM for 5 min in a zirconia ball mill (Retsch mixer mill MM301) with 3.0 mL of ethanol. Then 622.0 mg of Na-SAP110 suspended in 3 mL of ethanol were added and ground for another 5 min. 

The obtained sample, named CTA_FL_SAP_LAG, was dried in an oven at 50 °C for 1 h, then abundantly washed with water and ethanol to remove the excess surfactant and neutral molecules. The amount of CTABr employed for the synthesis is four times higher with respect to the cation exchange capacity (CEC) [46,47] of the clay (CEC equal to 50.4 meq/100 g) and double the theoretical CEC (104 meq/100 g) to push the reaction equilibrium and obtain full exchange.

In addition, a physical mixture was prepared, mixing in a mortar with gentle grinding of the same amounts of Fluorene, CTABr and Na-SAP110, without the solvent (sample Phys-Mix) and therefore in conditions where intercalation is not expected to occur.

### 3.4. One-Pot (OP) Synthesis

To obtain comparable saponite materials a fixed H_2_O/Si molar ratio equal to 110 was employed also in the OP method. 

The OP synthesis was conducted by modifying the method proposed by Bisio et al. [20] by adding CTABr and fluorene directly to the synthesis gel. The molar ratios among the reactants were the same as used for the LAG method. So, H_2_O/Si molar ratio equal to 110 and Fluorene and CTABr in a molar ratio of 0.4:1, were mixed, ground in a mortar with 15 mL of isopropanol, inserted into the synthesis gel, aged for 3 h and transferred to an autoclave for hydrothermal crystallization at 240 °C for 72 h. The obtained solid, named CTA-FL-SAP_OP, was finally filtered and washed with ethanol and water to remove the excess surfactant and fluorene not intercalated in the interlayer of saponite.

### 3.5. Deintercalation Procedure

To assess the presence of fluorene and of possible by-products in the CTA_FL_SAP_OP sample, an extraction with toluene using a Soxhlet extractor was performed. The toluene fraction was then treated with water to separate the water-soluble surfactant molecules (CTABr) from the fluorene and fluorene derivates and dried under vacuum. After solvent evaporation, a yellow slurry (Ext) was obtained. The saponite sample after the extraction procedure was named CTA_FL_SAP_OP_ext.

### 3.6. X-ray Powder Diffraction (XRPD)

The XRD patterns were obtained on a ARL XTRA48 diffractometer (Thermo Fisher Scientific, Waltham, MA, USA) using Cu Kα radiation (λ = 1.54062 Å). All powder diffraction patterns were measured in continuous mode using the following conditions: 2θ angular range 2–65° for standard measurements; tube power 45kV and 40mA, step size 0.02° 2θ. 

### 3.7. Scanning Electron Microscopy (SEM)

SEM images at different magnifications were recorded on a Quanta 200 FEI Scanning Electron Microscope equipped with EDAX EDS attachment(FELMI-ZFE, Graz, Austria), using a tungsten filament as the electron source at 25 KeV. The samples were differently prepared for the stub deposition. A small amount of ground and freeze-dried OP sample was dispersed by sonication in ethanol for 40 min. Then this suspension was dripped on a hot stub without adhesive tape and the surface was metalized with 30 nm of Au. While the LAG sample was ground and directly deposed on stub with adhesive tape and coated with 45 nm of Au. 

### 3.8. FTIR Spectroscopy

IR analyses were performed on a Fourier transform infrared (FTIR) Nicolet 5700 spectrometer (Thermo Optics)(Thermo Fisher Scientific, Waltham, MA, USA) at a resolution of 4 cm^−1^ in the spectral range from 4000 to 400 cm^−1^ and 128 scans. The samples were ground in a KBr pellet using a sample/KBr weight ratio of 1:10.

### 3.9. UV-Visible Diffuse Reflectance

Spectra were recorded using a Lambda 900-Perkin Elmer spectrophotometer (PerkinElmer, Waltham, MA, USA) in the spectral range from 190 to 800 nm. The samples were dispersed in a weight ratio of 1:20 with Barium Sulphate to attenuate the absorption. The Kubelka–Munk transformation was then applied to the raw data.

### 3.10. Raman Spectroscopy

Raman spectra were recorded on a Fourier-transformed Bruker RFS100 spectrophotometer (Bruker, Billerica, MA, USA), equipped with a Nd:YAG laser, emitting at 1.064 nm (NIR region), as the excitation source, and a liquid-nitrogen cooled Ge detector. The instrumental resolution was set at 4 cm^−1^ in the range of Raman shift 3500–200 cm^−1^ and 500 scans were acquired. 

### 3.11. Fluorescence Spectroscopy

Steady-state emission spectra (Horiba Ltd., Tokio, Japan) were recorded on a Horiba Jobin Yvon Model IBH FL-322 Fluorolog 3 spectrometer implemented with a 450-W xenon arc lamp, double-grating excitation and emission monochromators (2.1 nm/mm dispersion; 1200 grooves/mm), and a Hamamatsu Model R928 photomultiplier tube. Emission and excitation spectra were corrected for source intensity and emission spectral response by standard correction curves.

### 3.12. Thermogravimetric Measurements (TGA) and TGA-GC-MS Analyses

Thermogravimetric analyses under air were performed on a Setaram SETSYS Evolution instrument (gas flow 20 mL/min), heating the samples from 50 to 800 °C with a rate of 10 °C/min. Thermograms were corrected by subtraction of the background curve. 

Thermogravimetric analysis under inert gas and TGA-GC-MS analysis were performed using a Mettler TGA/SDTA 851e (Mettler Toledo, Columbus, OH, USA) purged with a steady flow of helium (inert gas) at a scanning rate of 10 °C/min from room temperature to 700 °C. Then, 1 mg of each sample was placed in an open alumina crucible. The GC-MS analysis was performed using a FINNIGAN TRACE GC-ULTRA and TRACE DSQ (Thermo Fisher Scientific, Waltham, MA, USA). The GC separation was carried out using a Phenomenex DB5-5ms capillary column (30 m, 0.25 mm i.d., 0.25 mm thickness). The injector temperature was set at 250 °C in the splitless mode and helium was used as carrier gas at a constant flow of 1.0 mL/min. The MS transfer line and the oven temperatures were set at 270 and 200 °C, respectively. The evolved gas from TGA was transferred to the GC-MS using an interface described in detail by Gianotti et al. [26] The heated transfer line (HTL1) from TGA to an automatic gas sampling system (AI) system, and the second heated transfer line (HTL2) from the AI to the GC-MS injector port were at 250 °C, AI temperature at 200 °C. The sampling frequency was 30 s^−1^. The sampled gas from the loop to the waste was switched after 10 s and the capacity of the injection loop was 2.5 mL. The MS signal was acquired in EI+ mode with an ionization energy of 70.0 eV and at the ion source temperature of 250 °C. The acquisition was performed both in full-scan mode, in the 20–350 *m*/*z* range and in Single Ion Monitoring (SIM) mode by acquiring the signals corresponding to CTABr at *m*/*z* = 95, fluorene at *m*/*z* = 166, fluorenone at *m*/*z* = 180 and the alkyl fluorene derivatives at *m*/*z*= 208 and 250. On the Na-SAP110, the survey of all the *m*/*z* fragments investigated was performed to obtain the background due to the inorganic matrix. The identification of the evolved products was performed by comparison of the retention times and spectra of commercial standards analyzed in the same experimental conditions.

## 4. Conclusions

A TGA-GC-MS method was proposed as a facile and direct approach to investigate the intercalation of organic guests into layered materials and to differentiate among the intercalated and surface-adsorbed organic molecules in critical cases not analyzable by other techniques. XRPD is the go-to technique to rapidly assess the intercalation of a guest into the inorganic host showing marked swelling, thus measuring the widening of the interlayer spacing. However, XRPD is less useful in cases such as saponite where the swelling is limited and, more importantly, is neither able to discriminate the presence of a guest compound adsorbed on the surface of the material and into defects, nor to assess the nature of the intercalated molecule when the co-intercalation of two different species such as CTABr and fluorene is attempted.

SEM characterization allowed to confirm that the synthesized materials retained their layered morphology. FTIR was able to give information on the surroundings of the guest molecules and allowed inferring the density and interaction between the CTA^+^ chains but fluorene signals were below its sensibility. The presence and dispersion of fluorene in the samples were confirmed by fluorescence spectroscopy thanks to the photoactivity of this compound. TGA-GC-MS was not only able to give information on the stability of the guest inside the layers but thanks to the MS probe and its high sensibility was able to detect and identify fluorene. Moreover, the presence of fluorene by-products was hinted at by spectroscopy and assessed after the extraction of the intercalates. This procedure was followed by UV-Vis, fluorescence and Raman spectroscopic characterization to demonstrate the presence of fluorenone as a by-product. This result was confirmed by TGA-GC-MS which, opposite to fluorescence spectroscopy, can be applied even with not photoactive co-intercalates. Moreover, TGA-GC-MS allowed confirming that the intercalation process of a guest into layered materials enhances the thermal stability of the obtained materials. 

Concerning the synthesis of co-intercalated organo-modified clays, both methods, LAG and one-pot, allowed the facile and fast intercalation of fluorene in saponite and can be used also for insertion of neutral compound in saponite by co-dispersing it with a cationic surfactant. However, fluorene by-products were obtained during the one-pot synthesis, decreasing its efficiency and applicability. Conversely, the LAG method is less aggressive and faster but despite the higher apparent loading of guest molecules, not all the guest is intercalated inside the layers, but a certain amount is adsorbed on the surface. The choice of the method should therefore be performed on the basis of the stability/reactivity of the guest and the importance of a high yield intercalation.

## Figures and Tables

**Figure 1 molecules-27-03048-f001:**
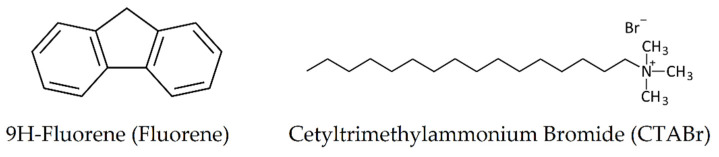
Molecular formulas of used chemical compounds.

**Figure 2 molecules-27-03048-f002:**
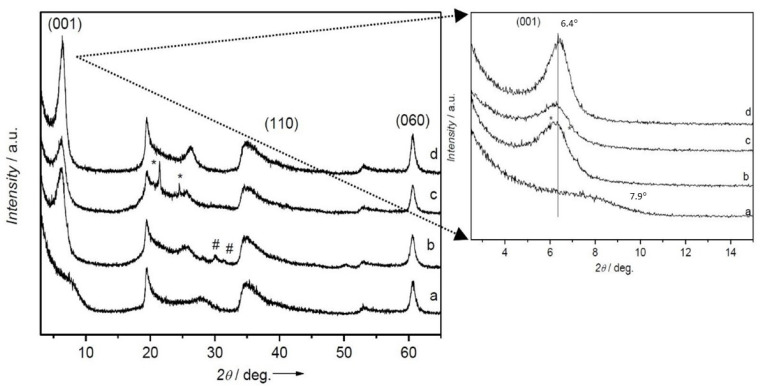
XRPD patterns recorded from 3° to 65° in 2*θ* range for samples (a) Na-SAP110, (b) CTA_FL_SAP_LAG, (c) CTA_SAP_LAG and (d) CTA_FL_SAP_OP. The peaks labeled with a star (*) are ascribed to NaBr formed during the cation exchange and those labeled with a number sign (#) to residual crystalline CTABr.

**Figure 3 molecules-27-03048-f003:**
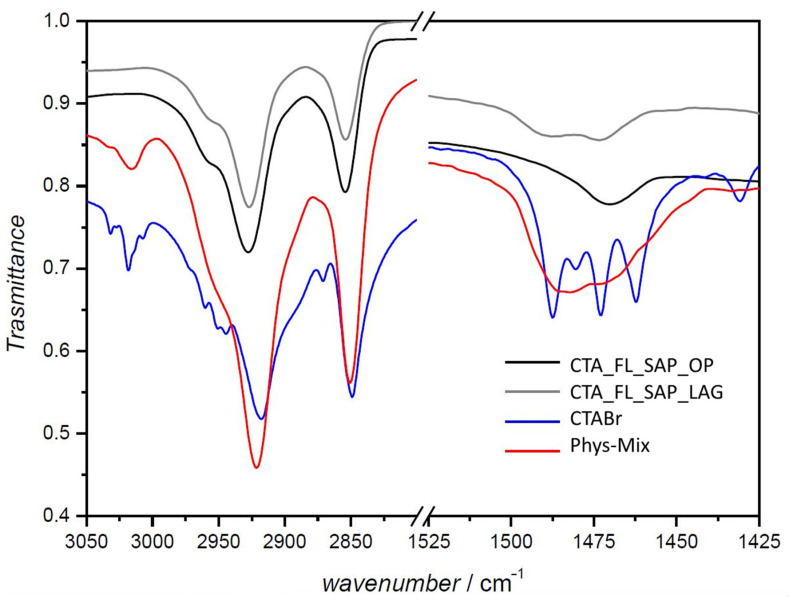
FTIR spectra of CTA_FL_SAP_OP (black), CTA_FL_SAP_LAG (grey), CTABr (blue), Phys-Mix (red). (Left) region of CH_2_ stretching modes; (right) region of CH_2_ bending modes.

**Figure 4 molecules-27-03048-f004:**
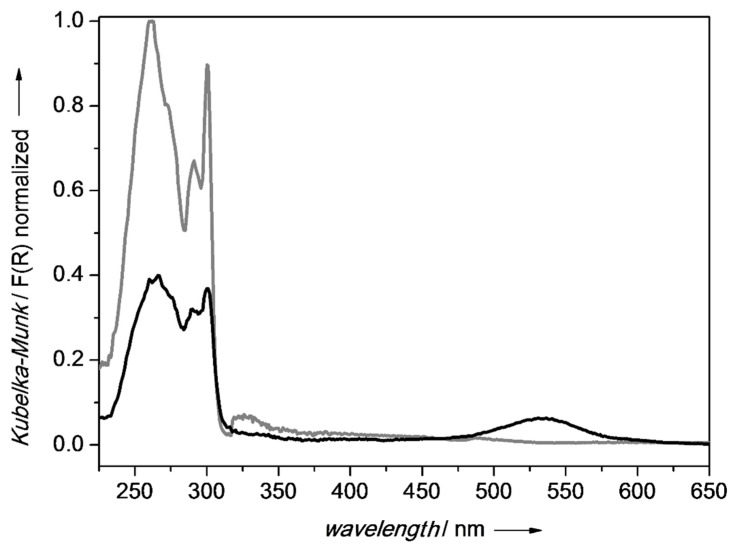
DR-UV/Vis spectra of CTA_FL_SAP_OP (black line) and CTA_FL_SAP_LAG (grey line) recorded from 200–800 nm.

**Figure 5 molecules-27-03048-f005:**
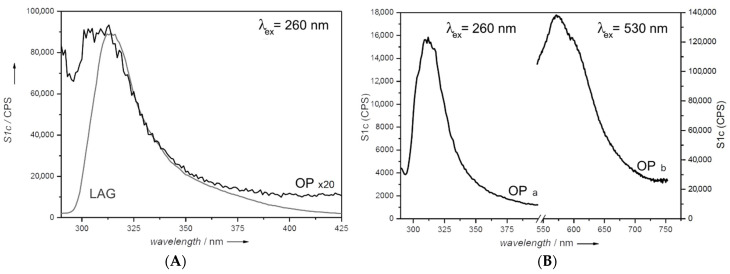
Fluorescence emission spectra. Left side (**A**), CTA_FL_SAP_OP (black curve) and CTA_FL_SAP_LAG (grey curve) recorded with an excitation λ of 260 nm (2.5 nm and 3 nm aperture slits and 2 s of integration time). The intensity of the black curve is multiplied 20 times. Right side (**B**), emission spectra of the CTA_FL_SAP_OP sample recorded with an excitation λ of 260 nm (curve a) and λ 530 nm (curve b) (both curves were recorded with 4 nm and 4 nm aperture slits and 2 s of integration time).

**Figure 6 molecules-27-03048-f006:**
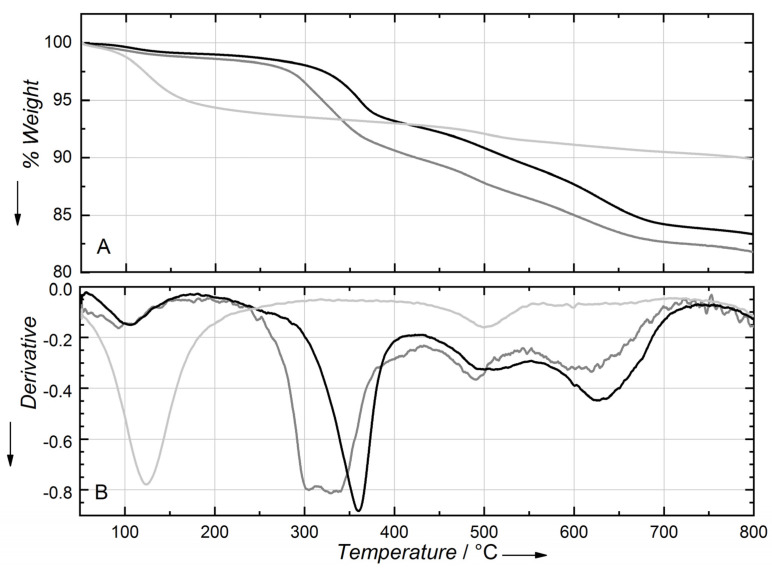
(**A**) TGA curves collected under airflow of CTA_FL_SAP_OP (black line) and CTA_Fl_Sap_LAG (grey line) and Na-SAP110 (light grey line). (**B**) The differential TG curves of the same samples.

**Figure 7 molecules-27-03048-f007:**
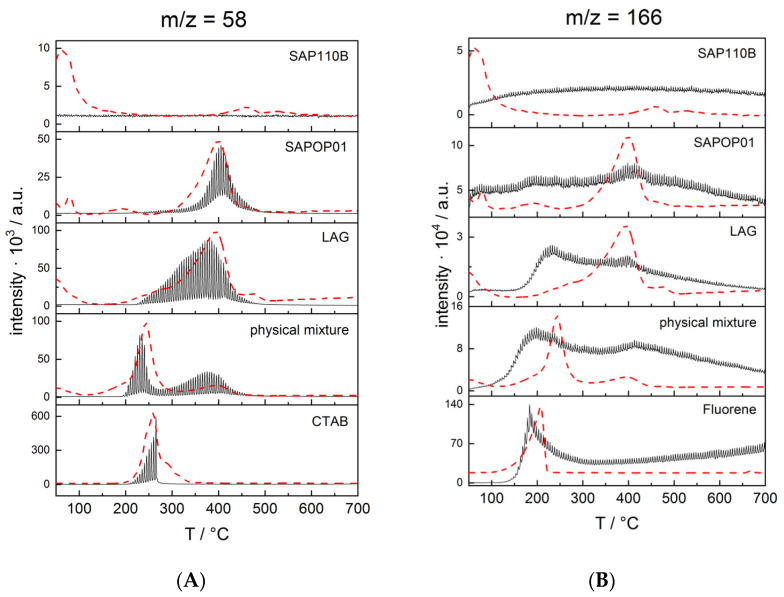
Evolution profiles of the signals at 95 *m*/*z* (**A**) and 166 m/z (**B**) recorded in the analyses of the pure samples (CTABr, Fluorene and Na-SAP110) and of the hybrid materials (CTA_FL_SAP_LAG and CTA_Sap_LAG). For comparison, the respective dTA curves are reported in dotted lines.

**Figure 8 molecules-27-03048-f008:**
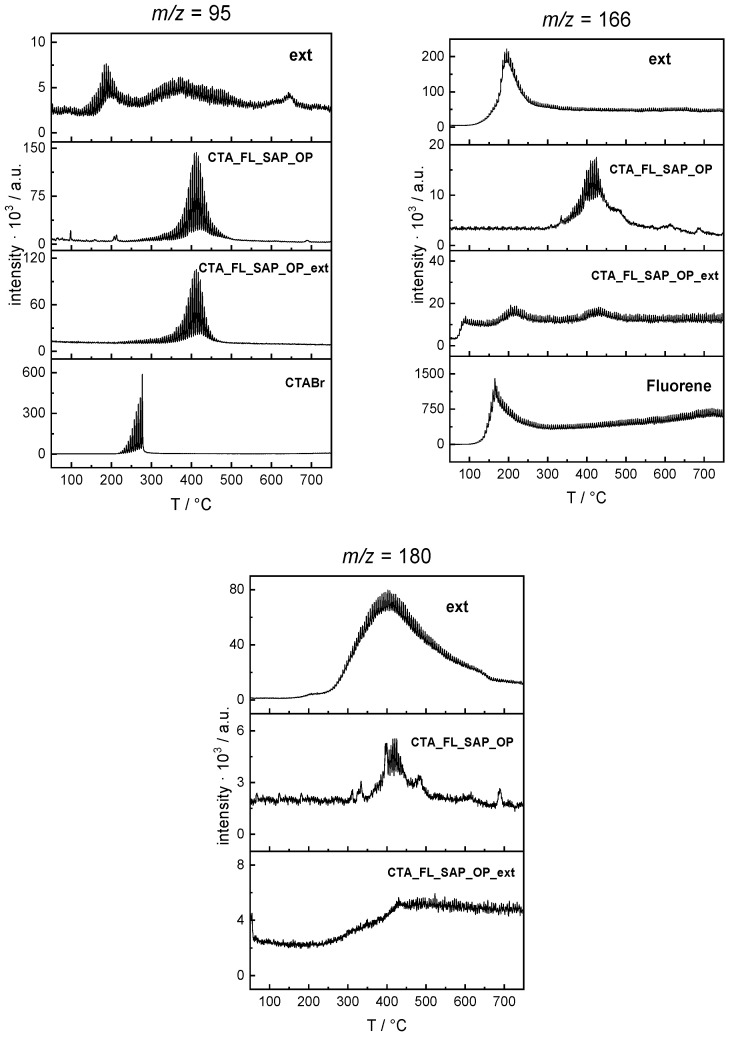
Evolution profiles of the signal at 95 *m*/*z* (CTABr) 166 *m*/*z* (Fluorene) and 180 *m*/*z* (fluorenone) recorded in the analyses of the pure samples (CTABr, Fluorene) and of the OP materials before and after the extraction together with the profile of the extract.

**Figure 9 molecules-27-03048-f009:**
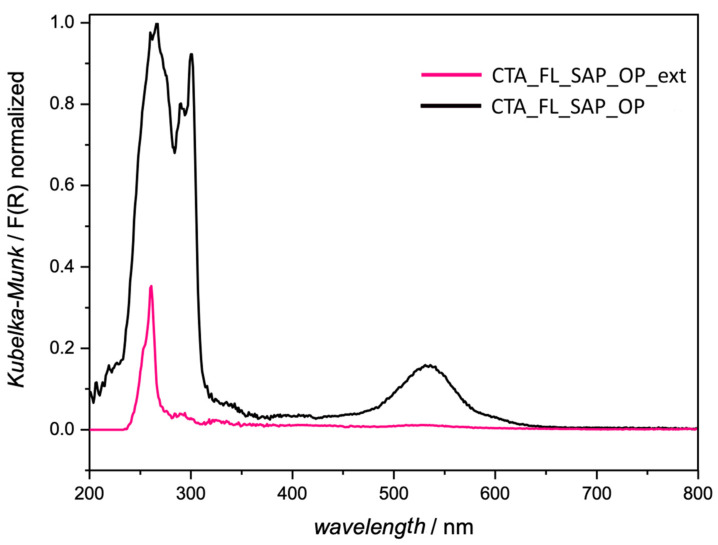
Normalized DR-UV/Vis spectra of CTA_FL_SAP_OP (black) and CTA_FL_SAP_OP_ext (pink).

**Figure 10 molecules-27-03048-f010:**
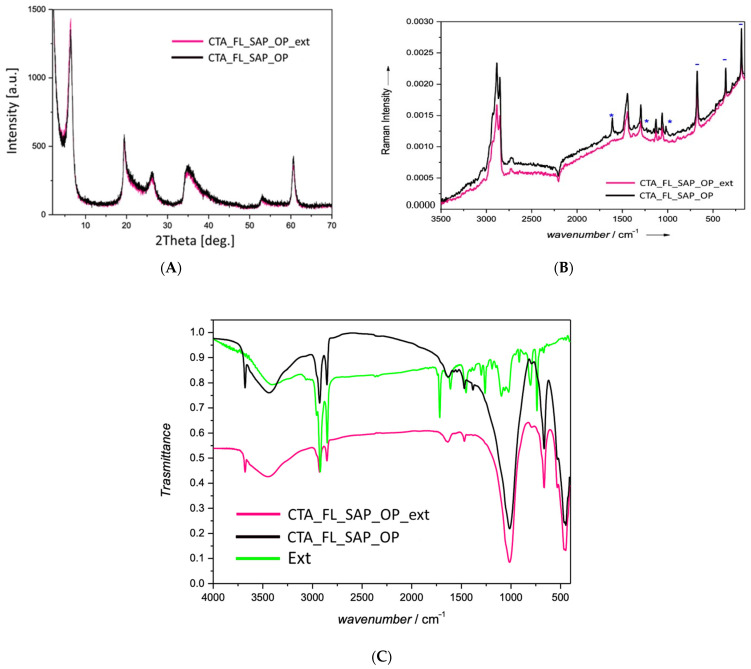
(**A**) XRPD pattern of the CTA_FL_Sap_OP sample before and after the extraction procedure, (**B**) Raman spectra of the CTA_FL_Sap_OP sample before and after the extraction procedure and (**C**) FTIR spectra of the CTA_FL_Sap_OP sample before, after the extraction procedure and of the extracted slurry (Ext).

## Data Availability

Data is contained within the article or supplementary material.

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
