# Peer review of "Analytical Characterization of the Intercalation of Neutral Molecules into Saponite"

_molecules, 2022, doi:10.3390/molecules27103048_

Round 1

Reviewer 1 Report

please check the objectives and the conclusion of this paper are not alligned.

the font size of the name for the molecular Figure 1 was not follow the journal format

please send the manuscript for proof read

Author Response

Dear Editor,

We are returning the manuscript after revision according to Reviewers’ comments. First of all, we would like to sincerely thank the Reviewers for the general appreciation of our manuscript and for their helpful suggestions.Please find below the detailed answers to Reviewers including the list of changes in the manuscript.

On behalf of all the authors,

Sincerely yours,

Eleonora Conterosito

Reviewer 1

please check the objectives and the conclusion of this paper are not alligned.

Answer: A paragraph was added in the conclusions.

the font size of the name for the molecular Figure 1 was not follow the journal format

Answer: we changed the font of the names inside the picture to match the article format but the size depends on the figure size/resolution

Reviewer 2 Report

Some comments and questions.

Fig.2. What is labeled with «#»?

Fig. 4 and 5. It would be highly desirable to provide data for the original components (Na-SAP110 first of all)

Fig. 6. As can be seen from the TG and DTG curves, the mass loss did not end at 800C. What was the reason for the measurement up to this temperature?

Line 280. What is the reason for the presence of signal 166 for the Na-SAP110?

What experimental data for sure indicate the presence of fluorene in the interlayer space, and not on the surface of the samples (taking into account the similarity of the MS 166 graphs for the physical mixture)?

There are some confusing sentences to correct.

Line 281 the description refers to the «profiles of the signal at 95 m/z» while in the captions in the figure 58 and 166 m/z…

Line 285. «As can be seen in Figure 5 a and b lower part» probably should be in Figure 7, but there is no «a» and «b» in Figure 7.

Line 286 «chosen fragments (95 m/z for CTABr and 166 m/z for fluorene)». In the captions in the figure, instead of it, 58 and 166 m/z…

Line 324 «As can be seen in the lower part of Figure 6, a signal deriving from the fragment at 180 m/z was found».  Figure 6 shows only TG and DTG data, not MS.

Author Response

Dear Editor,

We are returning the manuscript after revision according to Reviewers’ comments. First of all, we would like to sincerely thank the Reviewers for the general appreciation of our manuscript and for their helpful suggestions.Please find below the detailed answers to Reviewers including the list of changes in the manuscript.

On behalf of all the authors,

Sincerely yours,

Eleonora Conterosito

Reviewer 2

Some comments and questions.

Fig.2. What is labeled with «#»? 

Answer: The # marks residual crystalline CTABr, the label was added in the caption

Fig. 4 and 5. It would be highly desirable to provide data for the original components (Na-SAP110 first of all)

Answer: We added the UV/Vis spectra of Na-SAP110 and of fluorene in figure SI4 in the esi file. UV/Vis and photoluminescence spectra of CTABr can be found in the literature and this reference was added:  Materials 2020, 13, 2591; doi:10.3390/ma13112591. Fluorene emission is quenched at the solid state therefore the emission spectra was not recorded.

Fig. 6. As can be seen from the TG and DTG curves, the mass loss did not end at 800C. What was the reason for the measurement up to this temperature?

Answer: The measurement ended at 800 °C because we were interested in the degradation of the organic part. In an oxidizing atmosphere part of the organic remains longer inside the layers as carbonaceous residues that are in part removed at higher temperatures but their complete combustion in this case wouldn’t have added useful information.

Line 280. What is the reason for the presence of signal 166 for the Na-SAP110?

Answer: The 166 or the Na-SAP110 profile is not perfectly flat but no peaks or bands are evident.The  drift is due to the background corresponding to temperature increase of the TGA thermal gradient. It  is not unusual as seen for instance for fluorene alone before 400°C where no losses are present. A comment was added to better explain this behavior.

What experimental data for sure indicate the presence of fluorene in the interlayer space, and not on the surface of the samples (taking into account the similarity of the MS 166 graphs for the physical mixture)?

Answer: The bullet proof of the intercalation of organic molecules within disordered layered hosts (as saponite) is often a tough task. Typically, the intercalation is demonstrated by a combination of complementary techniques. In our case, TGA-GC-MS and FTIR were used. At first, the presence of two distinct TGA losses at different temperatures indicate two different environments for the fluorene. Fluorene released at lower temperatures can be explained as adsorbed on the surface, while the portion released at higher temperature is more tightly bound and thus most probably coming from inside the layers is stabilized and therefore. Both the TGA-GC-MS signals of CTABr and Fluorene, and FTIR suggest that intercalation has already happened to some degree. The demonstration of the insertion by appearance of a TGA or better by TGA-GC-MS peak at higher temperature is typically accepted in the literature. Moreover, the extraction procedure wasn't able to completely remove the fluorene indicating the strong interaction with saponite in agreement with successful intercalation. We agree that the bulletproof to be sure would be a single crystal X-ray structure for instance, absolutely unfeasible with observed low order powder morphology. A paragraph and some references were added in the manuscript.

There are some confusing sentences to correct.

Line 281 the description refers to the «profiles of the signal at 95 m/z» while in the captions in the figure 58 and 166 m/z…

Answer: The figure label unfortunately was wrong. The typo was corrected.

Line 285. «As can be seen in Figure 5 a and b lower part» probably should be in Figure 7, but there is no «a» and «b» in Figure 7. 

Answer: Correct, it was indeed figure 7 and we added the “a” and “b”.

Line 286 «chosen fragments (95 m/z for CTABr and 166 m/z for fluorene)». In the captions in the figure, instead of it, 58 and 166 m/z…

Answer: As in line 281, the figure label unfortunately was wrong. The typo was corrected.

Line 324 «As can be seen in the lower part of Figure 6, a signal deriving from the fragment at 180 m/z was found».  Figure 6 shows only TG and DTG data, not MS.

Answer: The reference was corrected; the correct figure is number 8.

Reviewer 3 Report

Reviewer’s comments on the manuscript: Analytical characterization of the intercalation of neutral molecules into saponite written by  V. Toson, D. Antonioli, E. Boccaleri, M. Milanesio, V. Gianotti, E Conterosito

The presented manuscript presents two strategies to prepare a fully synthetic organo-modified saponite hosting neutral molecules: the classical one-pot (OP) and the Liquid Assisted Grinding (LAG) method. Two synthetic methods for the preparation of saponites with a cationic surfactant (CTABr) and a neutral chromophore (Fluorene) were tested and the obtained products were initially characterized with the standard techniques (XRPD, SEM, TGA, IR, UV-Vis, Fluorescence and Raman spectroscopy) as well as thermogravimetry coupled with gas chromatography and mass spectroscopy (TGA-GC-MS) which allowed to identify the species present in the sample and the kind of interaction with the host by distinguishing between intercalated and adsorbed on the surface.

The manuscript is in agreements with journal’s fields of interests. It is very interesting and well organized. I would like to underline that measured samples were analyzed using a wide range of analytical method. The obtained results are promising and clearly presented. Thus my suggestion is minor revision.

Reviewers comments and suggestions:

  • Abstract should underline the relevance of the presented studies. Could you please add some information about the possibility of the practical usage of the obtained results.
  • All manuscript: It's probably the template's fault, but often words are not divided correctly into syllables. Some examples: lines 27; 36
  • line 41 a space is missing.
  • line 65: Please change the figure caption into: “Molecular formulas of used chemical compounds.”.
  • All manuscript: Please use the same unit (I mean the same abbreviation) for 1 Ångström [Å]. Please see the misleading differences in lines 120 and 121.
  • lines 137-163: Please add some references.
  • All manuscript: please decade if you want to provide a space between the number and its unit or not, and use the chosen format in all manuscript.
  • lines: 216-220: Is there any other explanation? I would like to encourage the Authors to please rewrite this part of text to be more clear.
  • lines 261-313: I do appreciate this part of measurements. Well done.
  • line 422: Could you please explain why was such a molar ratio used?
  • line 460: Capital letter –Spectra.
  • line 516: allows or allowed.

Author Response

Dear Editor,

We are returning the manuscript after revision according to Reviewers’ comments. First of all, we would like to sincerely thank the Reviewers for the general appreciation of our manuscript and for their helpful suggestions.Please find below the detailed answers to Reviewers including the list of changes in the manuscript.

On behalf of all the authors,

Sincerely yours,

Eleonora Conterosito

Reviewer 3

The presented manuscript presents two strategies to prepare a fully synthetic organo-modified saponite hosting neutral molecules: the classical one-pot (OP) and the Liquid Assisted Grinding (LAG) method. Two synthetic methods for the preparation of saponites with a cationic surfactant (CTABr) and a neutral chromophore (Fluorene) were tested and the obtained products were initially characterized with the standard techniques (XRPD, SEM, TGA, IR, UV-Vis, Fluorescence and Raman spectroscopy) as well as thermogravimetry coupled with gas chromatography and mass spectroscopy (TGA-GC-MS) which allowed to identify the species present in the sample and the kind of interaction with the host by distinguishing between intercalated and adsorbed on the surface.

The manuscript is in agreements with journal’s fields of interests. It is very interesting and well organized. I would like to underline that measured samples were analyzed using a wide range of analytical method. The obtained results are promising and clearly presented. Thus, my suggestion is minor revision.

Reviewers comments and suggestions:

Abstract should underline the relevance of the presented studies. Could you please add some information about the possibility of the practical usage of the obtained results. 

Answer: A paragraph was added to the abstract.

All manuscript: It's probably the template's fault, but often words are not divided correctly into syllables. Some examples: lines 27; 36 

Answer:  Yes, the division is done automatically and we don't know how to change it, but it will fixed within the final publication editing.

line 41 a space is missing.

Answer: The space was added.

line 65: Please change the figure caption into: “Molecular formulas of used chemical compounds.”.

Answer: Done

All manuscript: Please use the same unit (I mean the same abbreviation) for 1 Ångström [Å]. Please see the misleading differences in lines 120 and 121.

Answer: The character was corrected

lines 137-163: Please add some references.

Answer: We added references 31-33

All manuscript: please decade if you want to provide a space between the number and its unit or not, and use the chosen format in all manuscript.

Answer: Done

lines: 216-220: Is there any other explanation? I would like to encourage the Authors to please rewrite this part of text to be clearer.

Answer: We examined other explanations such as the possibility of having some monodispersed fluorene emitting at one wavelength and aggregated emitting at a higher wavelength (Journal of Polymer Science: Part A: Polymer Chemistry, Vol. 47, 4215–4233 (2009)) but this hypothesis was ruled out because the presence of fluorenone was suggested by the C=O band in the Raman spectra and confirmed by TGA-GC-MS. We rewrote the paragraph to explain this issue more clearly.

lines 261-313: I do appreciate this part of measurements. Well done.

Answer: The authors thank the reviewer for the nice comment.

line 422: Could you please explain why was such a molar ratio used?

Answer: The 0.4:1 ratio between CTABr and Fluorene was tuned in a preliminary experimental phase starting from literature procedures (see: “Unusual Incorporation of Neutral and Low Water-Soluble Guest Molecules into Layered Double Hydroxides: The case of cucurbit[6 and 7]uril Inclusion Hosts” - dx.doi.org/10.1021/cm102962g). At first the amount of CTABr was chosen equal to the CEC of the saponite, then this amount was increased while keeping it under the solubility limit to favor the complete exchange. The use of HCl and of different alcohols was also investigated. The details of these preliminary experiments were not reported to focus the manuscript on the analytical characterization of the materials but a comment and additional references were added in the manuscript, to explain the origin of the final used preparation procedure.

line 460: Capital letter –Spectra.

Answer: Corrected 

line 516: allows or allowed.

Answer: Corrected

Round 2

Reviewer 3 Report

Dear Authors,

The manuscript is ready for publication.

Kind regards,

Reviewer